# Accessing gold p-acid reactivity under electrochemical anode oxidation (EAO) through oxidation relay

Shuyao Zhang[1], Jingwen Wei[1], Xiaohan Ye[1], Angel Perez[1] & Xiaodong Shi ®[1] ✉

The gold π-acid activation under electrochemical conditions is achieved. While EAO allows easy access to gold(III) intermediates over alternative chemical oxidation under mild conditions, the reported examples so far are limited to coupling reactions due to the rapid Au[III] reductive elimination. Using aryl hydrazine·HOTf salt as precursors, the π-activation reaction mode was realized through oxidation relay. Both alkene and alkyne di-functionalization were achieved with excellent functional group compatibility and regioselectivity, which extended the versatility and utility of electrochemical gold redox chemistry for future applications.

With the improved C–C multiple bond activation reactivity and signature rapid reductive elimination, gold redox chemistry has received tremendous attention over the past decade as a cutting-edge direction in gold catalysis with unique reactivity[1–5]. While interesting transformations have been developed, one general concern of gold redox catalysis is the requirement of strong oxidants, such as Selectfluor and PhI(OAc)₂, to facilitate the initial gold(I) oxidation, due to the high Au[I]/Au[III] redox potential (1.41 eV)[6,7]. To extend the reaction protocol, gold(I) oxidation strategies have been developed. Some representative strategies include the photo or based promoted diazonium activation[8–15], silver-assisted P, N-hemilabile ligand promoted aryliodide oxidative addition, and oxazinium and sulfonium cation initiated gold(I) oxidation[16,17] etc. The recent development of electrochemical anode oxidation (EAO) provided an interesting strategy for achieving metal oxidation under mild conditions without the requirements of strong chemical oxidants[18–24]. The challenge was the competing metal cation cathode reduction, which gave rapid catalyst quenching. Through the tuning of electrodes and reaction conditions, our group recently reported EAO gold redox catalysis for alkyne oxidative coupling with proton reduction (formation of H₂) as the cathode counter reaction[25] (Fig. 1A). Later, this EAO promoted gold redox chemistry was successfully extended to sp-sp² coupling under basic conditions with O₂ reduction as the cathode counter reaction[26].

While promoting direct C–C coupling using EAO mediated gold redox chemistry is interesting, arguably, the most unique reactivity of cationic gold catalysis over other metal cations is the excellent π-acid activation ability toward C-C multiple bonds (alkene, alkyne, and allene)[27–32]. In both examples mentioned above, no gold-cation π-acid reactivity was observed, even the cationic gold(III) intermediates are formed during the process (Fig. 1B). Therefore, success protocol in accessing gold π-acid reactivity under EAO conditions is not only interesting as fundamental catalysis research, but also opens opportunities for transformations by combining the gold π-activation ability and EAO gold redox chemistry with no interruption from the traditionally required external strong oxidants.

In this work, we report an example of alkene and alkyne π-acid activation under EAO promoted gold redox catalysis through arylhydrazine involved oxidation relay (Fig. 1C).

## Results and discussion

### Reaction discovery for gold catalyzed π-activation under EAO conditions

As discussed above, although strong oxidants, Selectfluor or PIDA, could give gold(I) oxidation, the need for a stoichiometric amount of strong oxidative reagents greatly limited the application of this method for practical synthesis, both for cost and substrate compatibility. Recently, photo or base-promoted diazonium salt activation has been developed to generate L-Au[III]-Ar intermediate upon reacting Au[I] with the in-situ formed aryl radical[8–15,33]. While with a lot of successes, this method is limited to EWG-modified arene due to the need of reactive aryl radical for gold(I) oxidation. Bourissou[34–38] and Patil[39–46] recently reported the ligand enabled gold(I) oxidative addition to aryl iodide with the assistance of silver salts. The hemilabile P,N- ligand was applied to facilitate the initial oxidative

[1]Department of Chemistry, University of South Florida, Tampa, FL, USA. ✉e-mail: xmshi@usf.edu

**A)** Electrochemical anodic oxidation (EAO): a green and mild oxidation condition

ref 25, 26

e-Chem: *Good for C-C coupling*

**B)** The π-activation mode: unique reactivity of [Au]⁺, but no example under EAO

C-C coupling only

*no gold π-acid chemistry under EAO yet*

**C)** *This work:* First gold π-acid activation under EAO through oxidation relay

*direct [L-Au^I-Ar] EAO, no π-acid activation*

> 95%          **0%**

*critical OTf⁻ anion*

*in-situ formed [L-Au^III-Ar] via oxidation relay*

excellent regio selectivity;
> 30 example; up to 90% yield

**Fig. 1 | Achieving π-activation via electrochemical gold redox catalysis. A** Electrochemical anodic oxidation enabled gold redox catalysis on C-C coupling reactions. **B** Proposed π-acid activation under electrochemical condition. **C** Successful gold π-acid activation under EAO through oxidation relay.

addition and substrates with electron rich arene could be tolerated under this condition. While this seminal work showed good reactivity for both cross-coupling and π-activation, the requirement of at least one equivalent of Ag⁺ salts limited the practical application of this method due to low atom economy and concern about functional group tolerability.

With the high efficiency to achieve metal cation oxidation, EAO has been successfully applied to many transition-metal catalysis under controllable cell potential and steady reaction rate. Clearly, the EAO promoted gold redox chemistry is very attractive as it tackled one main challenge in gold redox catalysis, Au^I oxidation under mild conditions. The problem, however, is the rapid gold cation reduction on the cathode as Au⁰, which is very easy to form under redox conditions. Therefore, to facilitate EAO-assisted gold redox catalysis, developing effective cathode counter reactions (over gold cation reduction) is essential. After careful screening of reaction parameters, our group recently discovered that proton hydrogenation could be used as the cathode counter reaction if conducting the reaction under acidic conditions. Based on this mechanistic insight, we reported the alkyne oxidative coupling under EAO conditions as shown in Fig. 2A.

With the success on oxidative diyne coupling, we tried to extend this EAO promoted gold redox chemistry into aryl oxidative coupling using ArB(OH)₂. However, under the similar acidic conditions (MeCN/HOAc solvents), no coupling products were obtained due to rapid L-Au^I-Ar protodeauration (Fig. 2B). Interestingly, after extensive condition screening, we discovered that basic conditions could prevent the protodeauration with O₂ reduction (forming HO⁻) as the cathode counter reaction. The aryl coupling was successfully achieved under this EAO conditions. It is important to note that these studies not only provided two practical conditions for gold catalyzed C-C coupling under the electrochemical conditions, but also revealed some important mechanistic information of gold redox catalysis under EAO conditions. Monitoring the reaction with ³¹P NMR clearly confirmed the

formation of PPh₃-Au-C (C = alkynal or aryl) as the gold(I) resting state under both acidic and basic conditions. Interestingly, the reaction of two different aryl boronic acid (*para-t*Bu and *para*-COOMe) mixtures (1:1) gave dominant homo-coupling of the more electron-rich substrates (from *para-t*Bu substrate). This result clearly suggested that, while the oxidation of L-Au^I-Ar is the turnover limiting step, Au^I-Ar gave much faster transmetallation over Au^III-Ar. With these mechanistic insights, we put our attention on the much more challenging gold π-acid reactivity under the EAO conditions, which has not been achieved so far in literature.

As shown in Fig. 3A, monitoring the reaction of alkene **1a** and arylboronic acid **2a** under the previous optimized EAO conditions (via ³¹P and ¹⁹F NMR) showed the formation of di-aryl coupling product **4aa** and phenol **4ab** (from arene oxidation), with no alkene conversion and no compound **3a** formation observed at all.

The formation of coupling product **4aa** confirmed the gold redox cycle under the EAO condition. However, the 100% recovery of alkene **1a** suggested no [L-Au^III-Ar] π-complex formation (no alkene activation) under this condition. Considering the [Au^I-Ar] could give fast transmetallation to the Au^III intermediate, which eventually gave the coupling product through rapid reductive elimination, we postulated that to access Au^III π-acid reactivity, minimizing the transmetallation on Au^III intermediate is crucial. Based on this analysis, an oxidation relay process was proposed as shown in Fig. 3B. The key to this design is the application of some aryl electrophile surrogate to directly form [Au^III-Ar] intermediate without going through [Au^I-Ar], with the hope of preventing the coupling reaction path by minimizing transmetallation to Au^III intermediate.

Lei group has recently reported an interesting sonogashira carbonylation reaction under electrochemical oxidative conditions[47]. In that process, the aryl hydrazines were used as the precursors to form aryl radical through oxidative denitrogenation. Encouraged by this process, we postulated that aryl hydrazines might be applied as the

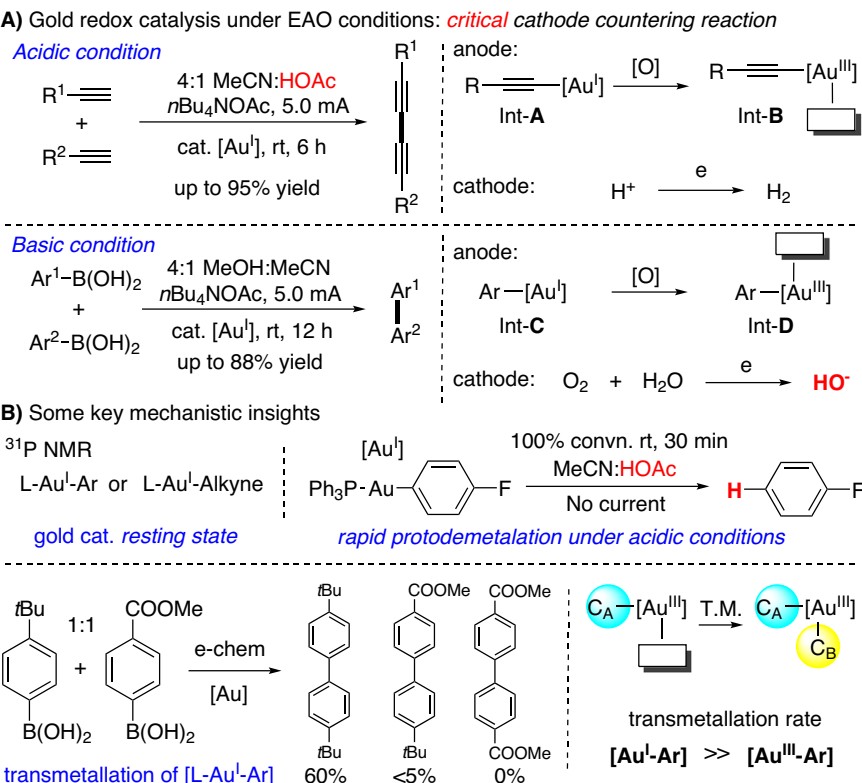

**Fig. 2 | Gold redox catalysis under electrochemical conditions. A** Gold redox catalysis under EAO conditions: critical cathode countering reaction. **B** Some key mechanistic insights.

aryl electrophile surrogates to oxidize gold catalyst through a similar electrochemical process. Considering that gold cation is known as carbophilic, either aryl radical or aryl cation could easily convert $Au^I$ into $Au^{III}$-Ar, especially under electrochemical oxidative conditions. To verify this hypothesis, aryl hydrazine **5a** was used to react with either terminal alkyne or aryl boronic acid under our previously developed EAO conditions. The corresponding coupling products were successfully observed, confirming the gold redox process under the EAO conditions with this hydrazine surrogate concept. It is important to note that for both coupling reactions, the homo-coupling $Ar^1$-$Ar^1$ is minor product, which is consistent with our hypothesis that the proposed oxidation relay could bypass the formation of $Au^I$–Ar intermediate (Fig. 3C).

During the preparation of this manuscript, Xie group reported a similar coupling process using aryl hydrazines[48]. Excellent reaction scope along with good functional group tolerability were achieved, highlighting the unique advantage of this EAO promoted gold redox process. However, despite the seminar works of identifying optimal conditions, 10% dppm(AuCl)$_2$, 1 eq. Phen, 3 eq. 2,6-di-$t$BuPy, MeCN, RVC/Ni, again, only coupling products were obtained with no π-acid activation observed even with substrates containing terminal alkyne and intramolecular NuH.

Encouraged by this result, to explore pi-acid reactivity based on our design, the reactions between alkene **1a** and hydrazine **5a** were performed under various EAO-promoted gold redox conditions. As shown in Table 1, the initial reaction gave diaryl coupling **4aa** as the major product and many un-identified byproducts. The desired alkene aminoarylation product **3a** was not observed at all, just like the results reported in recent Xie's work.

Considering that the cathode counter reaction was the reduction of $O_2$, the overall reaction was under basic condition due to the accumulated formation of $OH^-$ (monitoring the reaction pH confirmed the increasing of basicity over time). We postulate that the

coordination between nucleophiles and gold cation under basic conditions could block the coordination site and quench the π-acid activation. To avoid this undesired Nu-Au coordination, we performed the reaction under acidic conditions with the application of hydrazine salts ($ArNHNH_2$-HX), with the expectation to convert the cathode counter reaction into proton reduction (formation of $H_2$). Several hydrazine salts were prepared and applied into this EAO condition. Interestingly, while HCl and $HBF_4$ hydrazine salts gave almost no improvement, reaction with **5a**-HOTf gave the aminoarylation product **3a** in 16% isolated yield. This result is exciting as it presented a successful example of gold π-acid activation under EAO conditions. Further tuning the reaction parameters with various solvents, electrodes, electrolytes, and additives revealed the optimal conditions (10% $Ph_3PAuCl$, $nBu_4NPF_6$ as electrolytes and C/Pt as electrodes), giving the desired aminoarylation product **3a** in 92% isolated yields. Results from some representative alternative conditions are summarized in Table 2 (see detailed screening conditions in SI).

**Reaction optimization for EAO promoted gold π-activation**

As shown in Table 2, we first tested different MeCN and MeOH solvent mixture. The 10:1 ratio improved the yield to 32% (**entry 1-3**). Switching MeOH to other protic solvent led to a lower yield with more HAT product **4ad** (which was the major product of hydrazine oxidative decomposition, see **entry 12**). Aprotic solvents such as DCM cause the fast gold decomposition with $Au^0$ formation on the cathode with no formation of **3a** (**entry 5**). The $Ph_3P$ was identified as the optimal primary ligand, showing the good stability of $PPh_3AuCl$ under EAO conditions (monitoring the reaction with $^{31}P$ NMR, **entry 6**). Other tested ligands, such as other phosphine ligand and NHC carbene, gave rapid gold decomposition. Different electrolytes, such as $nBu_4NBF_4$ and $LiClO_4$, gave minimal influence on the reactivity (**entry 7**). Occasionally, when conducting the reaction under higher concentration (2X), slightly improved yield was observed (**entry 8**). Based on this

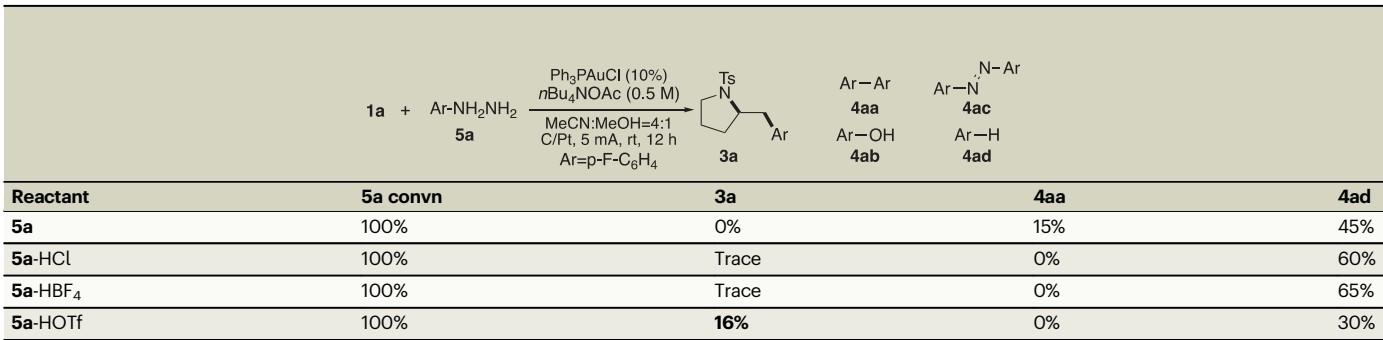

**Fig. 3 | EAO promoted gold redox catalysis through oxidation relay. A** Initial attempts for π-acid activation under EAO conditions. **B** By-passing [L-Au$^I$-Ar] to prevent side reactions. **C** Hydrazine serves as aryl electrophile "surrogate" for the initiation of oxidation relay.

## Table 1 | Gold π-activation under EAO conditions

| Reactant | 5a convn | 3a | 4aa | 4ad |
|---|---|---|---|---|
| **5a** | 100% | 0% | 15% | 45% |
| **5a**-HCl | 100% | Trace | 0% | 60% |
| **5a**-HBF$_4$ | 100% | Trace | 0% | 65% |
| **5a**-HOTf | 100% | **16%** | 0% | 30% |

condition screening, we concluded that the key factor influencing the yields of this reaction was the concentration of gold catalyst since the reaction turnover limiting step is likely the Au$^I$ oxidation to Au$^{III}$. When putting the gold catalyst at its maximum solubility (0.05 mmol in 5.5 mL solvent), we were pleased to obtain 70% yield of **3a** (**entry 10**). Finally, through the stepwise addition of hydrazines, the desired aminoarylation product **3a** was received in 92% isolated yield, with 100% conversion of **1a** (**entry 11**). The control experiments confirmed that both Au catalyst and current were necessary for this transformation (**entry 12-13**). It is important to note >90% of the gold catalyst can be recovered from the reaction mixture by flash column chromatography as Ph$_3$PAuCl, which suggested that the oxidation of hydrazine is the initiation of the oxidation relay. With the optimal conditions developed, we explored the reaction scope. The results are summarized in Fig. 4.

### Substrate scope screening

As shown in the reaction scope exploration, the monosubstituted alkene substrates worked well for this transformation. Both OH (alcohol and carboxylic acid) and NH (tosyl amine) could serve as

effective nucleophile for this transformation, giving the oxyarylation and aminoarylation correspondingly with good yields (**3a-3c**). For different ring size, both *5-exo-trig* and *6-exo-trig* cyclization were achieved with good reactivity. No *7-endo-trig* cyclization product was observed (see **3d** and **3e**). The 1,2-disubstituted E-alkene gave no cyclization with alkene remaining unreacted. This is likely due to the steric hindrance associated with these substrates that reduced the gold(III) coordination and hindered the effective π-activation. Various aryl hydrazines were also tested. To our satisfaction, both EDG and EWG-modified aryl hydrazines gave the desired oxy- and amino-arylation products in good yields (**3f-3o**). Notably, the benzylic proton is usually reactive under the EAO conditions due to the formation of reactive benzylic radical. To our surprise, the methylphenyl substituted hydrazine works fine under this condition, giving the desired product **3i**, though in a lower yield (likely due to the sequential product decomposition). This result suggested hydrazine is the more reactive reductant under this EAO condition and sequential oxidation is plausible with sequential oxidation sequence. The heterocycle substituted 4-pyridinylhydrazine gave no alkene conversion likely due to the rapid decomposition of the in-situ formed pyridinyl radical. Similarly, alkyl

**Table 2 | Optimization of reaction conditions[a, b]**

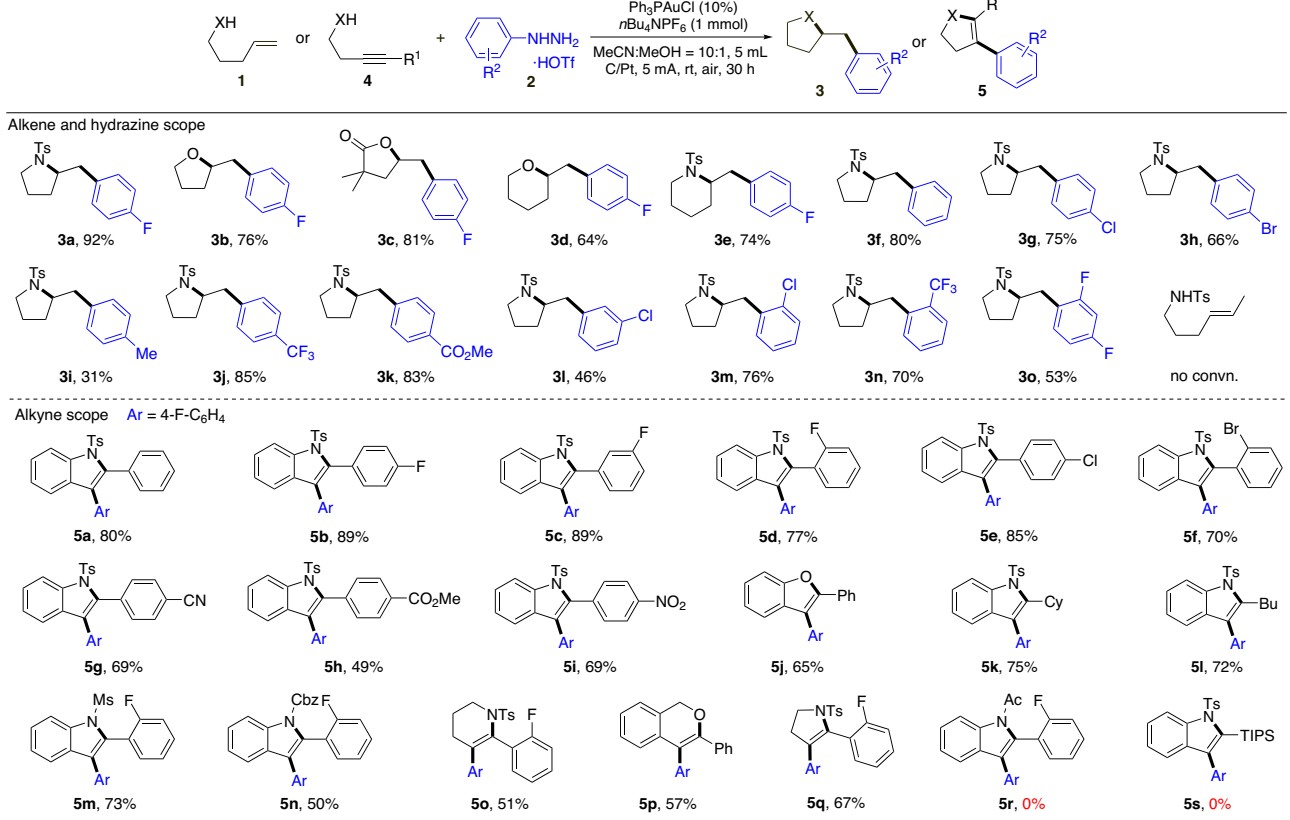

| Entry | Variation from "standard conditions" | convn. (2a) | 3a[c] | 4ad[c] |
|---|---|---|---|---|
| 1 | none | 100% | 32% | 29% |
| 2 | MeCN:MeOH = 4:1 | 100% | 16% | 30% |
| 3 | MeCN:MeOH = 1:1 | 100% | 12% | 26% |
| 4 | tBuOH instead of MeOH | 100% | 10% | 22% |
| 5 | DCM instead of MeOH | 100% | 0% | 48% |
| 6 | Other [Au] | 100% | <15% | Up to 50% |
| 7 | Other electrolytes | 100% | Up to 35% | Up to 40% |
| 8 | Double concentration (**1a, 2a**, [Au]) | 100% | 54% | 22% |
| 9 | 3 eq. **2a** | 100% | 40% | 27% |
| 10 | 0.5 mmol **1a**, 1.5 mmol **2a**, 0.05 mmol [Au] | 100% | 70% | 21% |
| **11** | 0.5 mmol **1a**, 1.5 mmol **2a**, 0.05 mmol [Au] (Add 0.5 mmol **2a**, run rxn for 10 h)*3 | 100% (100% for **1a**) | **92%** | 18% |
| 12 | No [Au] | 100% | 0% | 57% |
| 13 | No current | 0% | — | — |

[a]Conditions: **1a** (0.2 mmol), **2a** (0.4 mmol), Ph$_3$PAuCl (0.02 mmol), $n$Bu$_4$NPF$_6$ (1 mmol), MeCN (5 mL), MeOH (0.5 mL), C/Pt, 5 mA constant current, undivided cell, air, rt, 12 h.
[b]$^1$H NMR yields using 1,3,5-trimethoxybenzene as an internal standard.
[c]**3a** yield based on **1a**, **4ad** yield based on **2a**.

**Fig. 4 | Substrate scope for alkene, internal alkyne and hydrazine.** Conditions: **1** or **4** (0.5 mmol), **2** (1.5 mmol), Ph$_3$PAuCl (0.05 mmol), $n$Bu$_4$NPF$_6$ (1 mmol), MeCN (5 mL), MeOH (0.5 mL), C/Pt, 5 mA constant current, undivided cell, air, rt, add 0.5 mmol **2**, run rxn for 10 h, repeat 3 times. Isolated yields.

hydrazine and Tosyl hydrazine gave no alkene activation product due to the competing side reaction associated with the hydrazine decomposition under the EAO conditions (formation of reactive alkyl radical). The application of ArNHNH$_2$-HOTf salts was critical for this transformation as no alkene activation was observed in all these cases if only corresponding ArNHNH$_2$ was used for the reaction.

It is known that both Au$^I$ and Au$^{III}$ are good catalyst toward alkyne (C≡C) activation. One potential concern on the Au$^{III}$ promoted alkyne activation is the chemoselectivity between alkyne and resulting alkene. The lack of reactivity with more substituted alkene makes the chemoselective alkyne activation plausible under this EAO conditions. Various alkynes **4** were prepared and applied under this EAO mediated gold redox catalytic conditions (Fig. 4).

First, terminal alkyne gave cross coupling conjugated diyne product as previously reported. The internal alkynes work well under this EAO condition, giving the corresponding oxo- and amino-arylation products in good yields. The halogen substitution on arene showed little influence on this transformation, generating the desired products with good yields (**5b-5f**). This makes the sequential modification feasible from well-established aryl halide coupling. The EWG modified alkyne gave a negative impact on the reaction (**5g-5i**), likely due to reduced reactivity of the internal alkyne toward gold(III) π-activation. Impressively, although phenol is considered as a reactive functional group in anode oxidation, the phenol-modified alkyne gave product **5j** though with reduced yield. This result, once again, highlighted the unique advantage of electrochemical conditions with the controllable oxidation sequence. Impressively, aliphatic alkyne, despite containing less reactive C≡C, could also perform this transformation, giving the desired products (**5k, 5l**) in good yields. Interestingly, unlike the alkene activation, only *6-endo-dig* products (**5o, 5p**) were obtained as major products. Besides tosyl amine, other protecting groups, such as Ms amine and Cbz carbamide were also suitable nucleophiles for this transformation, allowing easier sequential modification/protection of the resulting products. Amide (Ac, **5r**) protecting group failed to give the desired product due to the rapid substrate decomposition (hydration under acidic conditions). Unfortunately, the TIPS modified internal alkyne remains unreacted under the optimal condition (**5 s**), which is likely associated with the less feasible nucleophilic addition containing this large sterically hindered modification. It is important to note that, under this EAO promoted gold redox condition, no alkyne hydration was observed in all cases, which highlighted the unique reactivity of this reaction protocol with many promising developments expected in the future. Overall, this electrochemical strategy presents a broad substrate scope for the alkene and alkyne difunctionalization, showing an alternative strategy for the synthesis of heterocycles with orthogonal reactivity from the typically known protocols.

In summary, we reported herein one successful example of the electrochemical promoted gold(I/III) redox chemistry for the alkene/alkyne π-activation. Arylhydrazine was applied as both coupling partner and radical precursor to promote gold oxidation. The acidic condition was critical for this process along with the application of proper anion. The success in achieving the π-acid reactivity of gold redox catalysis under electrochemical conditions not only provided a practical synthetic pathway to achieve the heterocycle derivatives under mild conditions, but also, more importantly, opened an opportunity to facilitate challenge transformations through this promising reaction path. Applications of this EAO-mediated gold redox π-acid activation strategy for other challenging transformations are currently undergoing in our lab.

## Methods

### General procedure for EAO promoted alkene/alkyne p-activation reaction

To a 10 mL ElectraSyn screwed vial with 387 mg $n$Bu$_4$NPF$_6$ (1 mmol) in MeCN:MeOH = 10:1 (5.5 mL), alkene **1** or alkyne 4 (0.5 mmol, 1.0 equiv.),

Ph$_3$PAuCl (0.05 mmol, 10 mol%) and the first batch of aryl hydrazine HOTf salt **2** (0.5 mmol, 1.0 equiv) was added. The vial was placed on IKA Carousel and run under constant current at 5 mA for 10 h. After the time was over, the cap was opened and another 1 eq hydrazine was added, then the reaction was performed under same condition for another 10 h. This step was repeated twice until all 3 eq hydrazine HOTf salt **2** was consumed. the solvent was removed under reduced pressure and the residue was purified by flash chromatography on silica gel to give desired product **3** or **5**.

## Data availability

The authors declare that the data supporting the findings of this study are available within the paper or its Supplementary Information files and from the corresponding author upon request.

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

## Acknowledgements

We are grateful to the NSF (CHE-1665122) and NIH (1R01GM120240-01) for financial support.

## Author contributions

S.Z., X.Y. and X.S. conceived the idea and co-wrote the manuscript with input from all of the co-authors. S.Z. conducted the majority of experiments and J.W. and A.P. assisted in the experiments. All authors contributed to discussions.

## Competing interests

The authors declare no competing interests.
