## [Peer Review File · Nature Communications]

REVIEWER COMMENTS

Reviewer #1 (Remarks to the Author):

In this publication, Shi and coworkers present a study on the utilization of electrochemical gold catalysis to achieve oxy-arylation and amino-arylation of alkenes and alkynes. This discovery is significant because it represents the first instance of gold's π -acid activation chemistry under electrochemical conditions. Undoubtedly, redox gold catalysis has emerged as a prominent research area for the cross-coupling and 1,2-difunctionalization reactions of C-C multiple bonds, which were previously inaccessible with Au(I) or Au(III) catalysis alone. However, due to the high oxidation potential and the complexities associated with the interconversion of Au(I) and Au(III) species in a catalytic manner, this process is more challenging compared to other transition metals (like Pd, Cu, or Ni). Traditionally, the utilization of super-stoichiometric external oxidants has been necessary to facilitate the Au(I)/Au(III) cycle. Unfortunately, these strong oxidants often display limited compatibility with various functional groups, thus restricting their applications. Later, various other approaches also have been developed in the past few years such as merged gold/photoredox or ligand-enabled gold redox catalysis, however, these approaches also suffer a few limitations. Interestingly, Shi and co-workers, in this report, provided a conceptually novel electrochemical approach to access Au(I)/Au(III) catalysis for the 1,2-difunctionalization of C-C multiple bonds. They have strategically designed the reaction and tackled the associated challenges of gold decomposition/reduction under electrochemical conditions. This work would be of great significance for further developments in the highly underdeveloped electrochemical Au(I)/Au(III) catalysis.

The manuscript is of great interest from both organometallic and synthetic viewpoints, and therefore, I strongly support its publication in this journal. The scope of the reaction has been explored thoroughly. The manuscript is easy and pleasant to read. In my view, there are only a few minor points worth considering before publishing these results:

- a) Change “n” to “n” in all tetrabutylammonium-based electrolytes; “tBuOH” to “tBuOH” and similar other letters in substrate scope. Maintain this uniformity throughout the manuscript.
- b) For optimization studies, the reaction was performed at 0.2 mmol scale ((limiting reagent - alkene)), and 1 mmol of electrolyte was used. For the 0.5 mmol scale (limiting reagent - alkene) also 1 mmol of electrolyte was used. Does an increase or decrease in the electrolyte concentration affect the outcome of the reaction? Also, it would be better to represent the additives' amount in terms of equivalent or molar concentration.
- c) In references, change “Acs catal” to “ACS Catal.” thoroughly.
- d) Page 2, Column 2: There is a mismatch for text citing Ref 41 and reference in the endnote. Please check.
- e) Page 1, Column 2: “Bourissou30-33 and Patil34-40 recently reported the ligand enabled gold(I) oxidative” correct the corresponding cited references. E.g. “Bourissou30-34 and Patil35-40 recently

reported----". Here, the authors could also cite recent examples of cross-coupling and 1,2-difunctionalization reactions utilizing this chemistry.

f) Page 1, Column 1: Change "electrides" to "electrodes".

g) Page 1, Column 2: Citing the Ref (Chem. Commun., 2018, 54, 11069) would be pertinent for the sentence "Recently, photo or base promoted diazonium salt activation has been ---- aryl radical."

h) P,N- ligand or P-N ligand, maintain uniformity throughout the manuscript.

i) Page 2, Column 2: "optimized EAO conditions (via NMR. 31P and 19F) showed" can be changed to "optimized EAO conditions (via 31P and 19F NMR) showed"

j) Page 3, Column 1: Change "to explore p-acid" to "to explore pi-acid"

k) Page 4, Column 1: Correct the Ph3P ligand in the sentence "The PPh3P was identified as the optimal primary ligand--"

l) Page 4: In conclusion, change "here in the first success examples of the electrochemical" to "here in the first successful examples of the electrochemical".

m) In ESI, Page 7: compounds 3a and 3b are racemic, please remove the configuration written in the text.

n) In ESI, all spectra should be labeled (type of spectrum and solvent).

o) In ESI, the optimization table, entry 16, showed that reducing the electrolyte concentration (0.5 mmol) helps increase the yield of the desired product. Do authors check this for 0.5 mmol (alkene) scale reactions?

Reviewer #2 (Remarks to the Author):

This work describes a significant advance in electrochemical gold catalysis by achieving the typical pi-philicity of gold species under EAO (electrochemical anode oxidation). The unique versatility of gold catalysis lies in the ability of gold catalyst in activating pi-systems, but this type of reactivity was not realized previously under EAO. This work represents a breakthrough. The use of arylhydrazinium salt as the source of aryl radical, despite also reported in a concurrent work by Xie (ref. #42), is thought to avoid excessive biaryl homodimer formation in the cases using arylboronic acids. This electrochemical strategy avoids the use of stoichiometric strong oxidants and is more sustainable and functional group-tolerant. This reviewer recommends its acceptance. A few issues, however, needs to be addressed. The discussion of the absence of LAu(I)Ar should be supported by NMR, as stated in Figure 2B. Despite this argument, Figure 3c shows that the dimer of arylhydrazine and the

dimer of arylboronic acid were formed in comparable amounts. How can this be explained in the context of arylhydrazine is more conducive to this work. In addition, there are a few grammatical/typos: 'condition' should be 'conditions', and 'sarrogate' should be 'surrogate'.

Reviewer #3 (Remarks to the Author):

This manuscript prepared by Xiaodong Shi and co-workers deals with a research on the electrochemical gold-catalyzed oxidative carboheterofunctionalization of alkenes. This reaction involves an anodic oxidation of aryl hydrazines to generate aryl radicals in situ, which combine with gold(I) complex to form L-Au(III)-Ar intermediates and trigger the following gold(III) participated cyclizations.

Although some new concepts/strategies and activation modes have been claimed by the authors, from the perspectives of mechanism or the transformation shown in this manuscript, this work seems routine. This reviewer thinks this manuscript is not important and is not suitable for Nat. Commun. Here are some points:

- 1) Similar types of carboheterofunctionalization of alkenes in oxidative gold catalysis with chemical oxidants and arylboronic acids had been reported more than 10 years ago (for a review on this field: Chem. Soc. Rev. 2021, 50, 10422–10450). For example, J. Am. Chem. Soc. 2010, 132, 1474–1475; Tetrahedron 2013, 69, 10375–10383, (not been cited in this manuscript). There is no obvious advantage with aryl hydrazines.
- 2) Electrochemical oxidation of aryl hydrazines to give aryl radicals has also been known, for example in ref. 42 and others.

In addition, this manuscript is not well prepared. For example:

- a) Fig. 1, 2 and 3 seem somehow duplicated, confusing and poorly organized. Readers cannot easily follow and get the points.
- b) The content on page 2 seems duplicated with the introduction part on page 1. There might not be necessary to illustrate/introduce the ideas of this work with 2 pages (the manuscript in total 5 pages including refs.)
- c) On page 2/column1/the last 5 lines, the authors said 'Interestingly, the reaction of two different aryl boronic acid (paratBu and para-COOMe) mixtures (1:1) gave dominate homo-coupling of the more electron-rich substrates (from para-tBu substrate). This result clearly suggested that, while the oxidation of L-AuI-Ar is the turnover limiting step, AuI-Ar gave much faster transmetallation over AuIII-Ar.' This reviewer does not see the logic 'clearly'.

d) In Fig. 1-C, 'critical anion' was used to probably show the importance of 'OTf-' to their reaction. If this is the case, here 'critical anion' might not be suitable to be added below the structure of hydrazine salt. This hydrazine salt is a species but not just an anion. In addition, here in the introduction part, nothing about this 'critical anion' was even mentioned on page 1.

e) On page 1/column 2/line 1, 'under EAO promoted gold redox catalysis through arylhydrazine assisted oxidation relay (Figure 1C).' Here, 'arylhydrazine assisted' might not be accurate, since it is involved in this reaction and is stoichiometrically consumed. Actually, the real species involved to combine/oxidize Au(I) is the aryl radical generated by the electrochemical oxidation of arylhydrazine, but not arylhydrazine itself.

f) On page 2/column 1/line11, 'Clearly, the EAO promoted gold redox chemistry is very attractive as it tackled the core challenges in gold redox catalysis, AuI oxidation under mild conditions' Here, 'it tackled the core challenges' might not be suitable and it might be too arbitrary to make this comment.

g) On page 2/column 2/line1, 'With these mechanistic insights, we put our attention on the much more challenging gold π -acid reactivity under the EAO conditions,' There seems no clear logical connection between the content of 'these mechanistic insights' on this paragraph and the idea of 'challenging gold π -acid reactivity under the EAO conditions'.

h) The two comments on page2/column2/paragraph3 and page4/column1/line23 seem contradictory: 'either aryl radical or aryl cation could easily convert AuI into AuIII-Ar,' and 'since the reaction turnover limiting step is likely the AuI oxidation to AuIII.'

i) On page4/column1/line13, 'PPh3P' is not correct and line 16, 'R3P' is not suitable, 'R' should be defined, otherwise meaningless.

j) On page4/column1/paragraph3, The authors said 'The 1,1 or 1,2-disubstituted alkenes (Z or E) gave no cyclization with alkene remaining unreacted (see details in SI).' and included a general structure of internal alkenes, noting 'no convn.' in Table 2. This reviewer checked the 'details in SI', and only one internal E-alkene was shown in 'Substrate scope of failed alkenes/hydrazines' on page 6 in SI. It might be too arbitrary to conclude that with the negative results obtained with 1 or several tries. More importantly, probably, this is scientifically and logically incorrect to conclude 'xx do not work' based on failed examples.

Revisions based on editor's and reviewers' comments:

Response to Reviewer Comment #1:

Comments to the Author:

In this publication, Shi and coworkers present a study on the utilization of electrochemical gold catalysis to achieve oxy-arylation and amino-arylation of alkenes and alkynes. This discovery is significant because it represents the first instance of gold's π -acid activation chemistry under electrochemical conditions. Undoubtedly, redox gold catalysis has emerged as a prominent research area for the cross-coupling and 1,2-difunctionalization reactions of C-C multiple bonds, which were previously inaccessible with Au(I) or Au(III) catalysis alone. However, due to the high oxidation potential and the complexities associated with the interconversion of Au(I) and Au(III) species in a catalytic manner, this process is more challenging compared to other transition metals (like Pd, Cu, or Ni). Traditionally, the utilization of super-stoichiometric external oxidants has been necessary to facilitate the Au(I)/Au(III) cycle. Unfortunately, these strong oxidants often display limited compatibility with various functional groups, thus restricting their applications. Later, various other approaches also have been developed in the past few years such as merged gold/photoredox or ligand-enabled gold redox catalysis, however, these approaches also suffer a few limitations. Interestingly, Shi and co-workers, in this report, provided a conceptually novel electrochemical approach to access Au(I)/Au(III) catalysis for the 1,2-difunctionalization of C-C multiple bonds. They have strategically designed the reaction and tackled the associated challenges of gold decomposition/reduction under electrochemical conditions. This work would be of great significance for further developments in the highly underdeveloped electrochemical Au(I)/Au(III) catalysis. The manuscript is of great interest from both organometallic and synthetic viewpoints, and therefore, I strongly support its publication in this journal. The scope of the reaction has been explored thoroughly. The manuscript is easy and pleasant to read. In my view, there are only a few minor points worth considering before publishing these results:

Thanks very much for reviewer's insightful comments and suggestions. It is always a great pleasure to have expert opinion in the reviewing process and we are extremely grateful for the reviewer in helping us improving the manuscript! We have revised the manuscripts as suggested. Please find our point-by-point response in the following.

a) Change "n" to "n" in all tetrabutylammonium-based electrolytes; "tBuOH" to "tBuOH" and similar other letters in substrate scope. Maintain this uniformity throughout the manuscript.

Thanks for the reviewer's suggestion. We have changed the format of all letters as suggested.

b) For optimization studies, the reaction was performed at 0.2 mmol scale (limiting reagent - alkene), and 1 mmol of electrolyte was used. For the 0.5 mmol scale (limiting reagent - alkene) also 1 mmol of electrolyte was used. Does an increase or decrease in the electrolyte concentration affect the outcome of the reaction? Also, it would be better to represent the additives' amount in terms of equivalent or molar concentration.

Thanks for the reviewer's suggestion. The electrolyte is not participating in the reaction. The concentration of electrolyte is mainly affecting the voltage under constant current, so it was not changed while we perform the reaction under different starting material concentrations. That's also why we did not represent the electrolyte's amount in terms of equivalent. We also tried different electrolyte concentration when optimizing conditions, but it has very little influence on this reaction (some conditions are summarized in SI).

c) In references, change "Acs catal" to "ACS Catal." thoroughly.

Thanks for the reviewer's suggestion. We have fixed this mistake as suggested.

d) Page 2, Column 2: There is a mismatch for text citing Ref 41 and reference in the endnote. Please check.

Thanks for the reviewer's comment. The reference is correct. We have changed the citing text to fit the reference.

e) Page 1, Column 2: "Bourissou30-33 and Patil34-40 recently reported the ligand enabled gold(I) oxidative" correct the corresponding cited references. E.g. "Bourissou30-34 and Patil35-40 recently reported----". Here, the authors could also cite recent examples of cross-coupling and 1,2-difunctionalization reactions utilizing this chemistry.

Thanks for the reviewer's suggestion. We have added the references as suggested.

f) Page 1, Column 1: Change "electrides" to "electrodes".

Thanks for the reviewer's suggestion. We have fixed this mistake as suggested.

g) Page 1, Column 2: Citing the Ref (Chem. Commun., 2018, 54, 11069) would be pertinent for the sentence "Recently, photo or base promoted diazonium salt activation has been ---- aryl radical."

Thanks for the reviewer's suggestion. We have added the reference as suggested.

h) P,N- ligand or P-N ligand, maintain uniformity throughout the manuscript.

Thanks for the reviewer's suggestion. We have changed all descriptions to "P,N- ligand".

i) Page 2, Column 2: "optimized EAO conditions (via NMR. 31P and 19F) showed" can be changed to "optimized EAO conditions (via 31P and 19F NMR) showed"

Thanks for the reviewer's suggestion. We have fixed the description as suggested.

j) Page 3, Column 1: Change "to explore p-acid" to "to explore pi-acid"

Thanks for the reviewer's suggestion. We have fixed this mistake as suggested.

k) Page 4, Column 1: Correct the Ph₃P ligand in the sentence “The PPh₃P was identified as the optimal primary ligand---”

Thanks for the reviewer’s suggestion. We have fixed this mistake as suggested.

l) Page 4: In conclusion, change “here in the first success examples of the electrochemical” to “here in the first successful examples of the electrochemical”.

Thanks for the reviewer’s suggestion. We have changed the text as suggested.

m) In ESI, Page 7: compounds 3a and 3b are racemic, please remove the configuration written in the text.

Thanks for the reviewer’s suggestion. We have fixed this mistake as suggested.

n) In ESI, all spectra should be labeled (type of spectrum and solvent).

Thanks for the reviewer’s suggestion. We have labeled all spectra as suggested.

o) In ESI, the optimization table, entry 16, showed that reducing the electrolyte concentration (0.5 mmol) helps increase the yield of the desired product. Do authors check this for 0.5 mmol (alkene) scale reactions?

Thanks for the reviewer’s comment. During our condition screening process, we figured out that the key factor that would influence the yield was the concentration of gold catalyst in the reaction system. The concentration of electrolyte, as described in question b), has very little influence on the outcome. Based on that result, we did not further check the different electrolyte concentration after we changed to 0.5 mmol (alkene) scale reactions.

Response to Reviewer Comment #2:

Comments to the Author:

This work describes a significant advance in electrochemical gold catalysis by achieving the typical pi-philicity of gold species under EAO (electrochemical anode oxidation). The unique versatility of gold catalysis lies in the ability of gold catalyst in activating pi-systems, but this type of reactivity was not realized previously under EAO. This work represents a breakthrough. The use of arylhydrazinium salt as the source of aryl radical, despite also reported in a concurrent work by Xie (ref. #42), is thought to avoid excessive biaryl homodimer formation in the cases using arylboronic acids. This electrochemical strategy avoids the use of stoichiometric strong oxidants and is more sustainable and functional group-tolerant. This reviewer recommends its acceptance. A few issues, however, needs to be addressed.

Thanks very much for reviewer’s insightful comments and suggestions. Again, we are grateful to reviewer in helping us improve the manuscript. We have revised the manuscripts as suggested. Please find our point-by-point response in the following.

a) The discussion of the absence of LAu(I)Ar should be supported by NMR, as stated in Figure 2B.

Thanks for the reviewer's comment. This part of work has been described in our previous electrochemistry gold catalysis paper. We're only using the result here to remind the key chemistry. Under the acidic condition, the gold catalyst will decompose and form a thin layer of purple precipitation within 30 minutes in the reaction vial. ^{19}F NMR confirmed that the products were all fluorobenzene, while no Ph_3PAuAr was left. The corresponding pictures of this decomposition are added in the revised SI for clarification.

b) Despite this argument, Figure 3c shows that the dimer of arylhydrazine and the dimer of arylboronic acid were formed in comparable amounts. How can this be explained in the context of arylhydrazine is more conducive to this work.

This is an excellent point. We thank for the reviewer for this insightful comment. The key point of reaction shown in Figure 3c is to testify whether aryl hydrazine could be applied as a coupling partner in electrochemical gold redox conditions. We have previously shown that under the standard reaction condition, homocoupling would happen if there is no Ar^1NHNH_2 . We're glad to see that comparable amount of cross-coupling products were generated with the addition of 1 eq. Ar^1NHNH_2 in both reactions, similar to Xie's recent Nat. Synth. paper. With the condition optimization, we successfully avoided the generation of homocoupling product $\text{Ar}^1\text{-Ar}^1$. This is actually the highlight of this work. We revised the figure for clarification as suggested. Thanks the reviewer again for the helpful comment.

c) In addition, there are a few grammatical/typos: 'condition' should be 'conditions', and 'sarrogate' should be 'surrogate'.

Thanks for the reviewer's suggestion. We have fixed these mistakes as suggested.

Response to Reviewer Comment #3:

Comments to the Author:

This manuscript prepared by Xiaodong Shi and co-workers deals with a research on the electrochemical gold-catalyzed oxidative carboheterofunctionalization of alkenes. This reaction involves an anodic oxidation of aryl hydrazines to generate aryl radicals in situ, which combine with gold(I) complex to form L-Au(III)-Ar intermediates and trigger the following gold(III) participated cyclizations.

Although some new concepts/strategies and activation modes have been claimed by the authors, from the perspectives of mechanism or the transformation shown in this manuscript, this work seems routine. This reviewer thinks this manuscript is not important and is not suitable for Nat. Commun. Here are some points:

1) Similar types of carboheterofunctionalization of alkenes in oxidative gold catalysis with chemical oxidants and arylboronic acids had been reported more than 10 years ago (for a review on this field: Chem. Soc. Rev. 2021, 50, 10422–10450). For example, J. Am. Chem. Soc. 2010, 132, 1474–1475; Tetrahedron 2013, 69, 10375–10383, (not been cited in this manuscript). There is no obvious advantage with aryl hydrazines.

First, we are grateful to the reviewer for spending time to review this work and provide opinions in helping us improve this manuscript!

However, with deep respect, it seems the reviewer missed the key point of the chemistry we intended to report here. The focus of this work is not for transformation, but for new chemistry development. As also pointed by reviewer #1 and #2, the electrochemical promoted gold redox chemistry is of general interest as it provides a much milder conditions in achieving gold oxidation and will initiate many potential transformations both for academic research and potentially practical industrial process. However, with the limited works reported so far, no success π -acid chemistry has been achieved so far. It is also well-known that gold cations is one of the most effective π -acid in promoting alkyne and alkene activation. It is critically important if practical conditions could be discovered/developed to facilitate this chemistry under the electrochemical conditions and that is what we report here. It is our common practice that to develop new chemistry, a well-documented transformation will be likely the most practical approach to gain new chemistry insight. Some examples in the community is that so many different catalytic systems have been developed in hydroamination, cross coupling with different metal and coupling partner etc.

With these new conditions developed, it is fully expected that the combination of this new π -acid protocol under E-chem conditions will be adopted by the researchers in the community for many new transformations to come by combining the gold π -acid chemistry and redox catalysis under the mild e-chem conditions.

2) Electrochemical oxidation of aryl hydrazines to give aryl radicals has also been known, for example in ref. 42 and others.

Yes, electrochemical oxidation of aryl hydrazine is known in literature, but application of this oxidation relay strategy to solve a critical gold redox chemistry with accessibility in π -acid activation is for the first time. Similarly, Pd catalyzed aryl iodide oxidative addition is well known in literature. However, the application of P-N ligand in producing gold oxidative addition to aryl-iodide by Bourissou and Patil is considered a breakthrough in gold redox chemistry with many new chemistry followed up later.

In addition, this manuscript is not well prepared. For example:

a) Fig. 1, 2 and 3 seem somehow duplicated, confusing and poorly organized. Readers cannot easily follow and get the points.

b) The content on page 2 seems duplicated with the introduction part on page 1. There might not be necessary to illustrate/introduce the ideas of this work with 2 pages (the manuscript in total 5 pages including refs.)

c) On page 2/column1/the last 5 lines, the authors said 'Interestingly, the reaction of two different aryl boronic acid (para-tBu and para-COOMe) mixtures (1:1) gave dominate homo-coupling of the more electron-rich substrates (from para-tBu substrate). This result clearly suggested that, while the oxidation of L-AuI-Ar is the turnover limiting step, AuI-Ar gave much faster transmetallation over AuIII-Ar.' This reviewer does not see the logic 'clearly'.

We respectfully accept the critics and carefully re-evaluated the manuscripts. However, it seems that this reviewer might lacking the familiarity of electrochemical mediated gold redox chemistry, which is also evidenced by above comments. The general logic flow are based on what has been reported in E-chem promoted gold redox catalysis. While there are only four papers reported so far, two by our group and the other two from Xie (Nat. Syn 2023) and Patil (JACS, 2023), a lot of background information have been discussed in those papers. It is our comment practice to not over-stated the simple background if they have been previously reported. This reviewer is clearly expert in methodology development. However, it seems that there is miss information/background on the e-chem gold redox history. More importantly, it seems that the other two reviewer have no problem in following the chemistry with clear understanding of the logic flow. We do make good efforts in revising figures and introduction to make sure our points getting accosted. Overall, we remain gratitude to this reviewer in pointing the potential issues and helping us improve the manuscript.

d) In Fig. 1-C, 'critical anion' was used to probably show the importance of 'OTf-' to their reaction. If this is the case, here 'critical anion' might not be suitable to be added below the structure of hydrazine salt. This hydrazine salt is a species but not just an anion. In addition, here in the introduction part, nothing about this 'critical anion' was even mentioned on page 1.

Thanks for the reviewer's suggestion. We have changed the word to "critical OTf anion" in Fig. 1 for clarification.

e) On page 1/column 2/line 1, 'under EAO promoted gold redox catalysis through arylhydrazine assisted oxidation relay (Figure 1C).' Here, 'arylhydrazine assisted' might not be accurate, since it is involved in this reaction and is stoichiometrically consumed. Actually, the real species involved to combine/oxidize Au(I) is the aryl radical generated by the electrochemical oxidation of arylhydrazine, but not arylhydrazine itself.

Thanks for the reviewer's suggestion. We have changed "assisted" to "involved" in the sentence.

f) On page 2/column 1/line11, 'Clearly, the EAO promoted gold redox chemistry is very attractive as it tackled the core challenges in gold redox catalysis, Au(I) oxidation under mild conditions' Here, 'it tackled the core challenges' might not be suitable and it might be too arbitrary to make this comment.

Thanks for the reviewer's suggestion. We have changed "the core challenges" to "one main challenge in gold redox catalysis" for clarification.

g) On page 2/column 2/line1, 'With these mechanistic insights, we put our attention on the much more challenging gold π -acid reactivity under the EAO conditions,' There seems no clear logical connection between the content of 'these mechanistic insights' on this paragraph and the idea of 'challenging gold π -acid reactivity under the EAO conditions'.

Thanks for the reviewer's comments. It seems that this is the research philosophic question: to come out a potential solution, what is the best way to do it? By trying different conditions

or rational design based on reaction mechanism? It is my understanding and daily practice to develop new chemistry based on mechanistic insights followed by rational design. Knowing gold catalysis, I am confident to say that accessing gold π -acid reactivity under the EAO conditions is one of the major challenges in current research. The rationale we presented here is that to get Au(III) π -acid reactivity, formation of L-Au(I)-Ar followed by oxidation is not valid due to the fast Au(III) reductive elimination (just as we stated in the manuscript).

h) The two comments on page2/column2/paragraph3 and page4/column1/line23 seem contradictory: 'either aryl radical or aryl cation could easily convert Au(I) into Au(III)-Ar,' and 'since the reaction turnover limiting step is likely the Au(I) oxidation to Au(III).'

Thanks for the reviewer's comment. The oxidation of Au(I) to Au(III) by aryl radical is easy, but the concentration of aryl radical in the reaction system is very low. Tracking the reaction with ^{31}P NMR also showed that most of the gold catalysts were remaining at Au(I) stage during the reaction process (resting state). Based on these evidences, we determined that the rds should be the oxidation of gold catalyst in this reaction and this is also well-documented in gold catalysis literatures.

i) On page4/column1/line13, 'PPh3P' is not correct and line 16, 'R3P' is not suitable, 'R' should be defined, otherwise meaningless.

Thanks for the reviewer's suggestion. We have fixed these mistakes as suggested.

j) On page4/column1/paragraph3, The authors said 'The 1,1 or 1,2-disubstituted alkenes (Z or E) gave no cyclization with alkene remaining unreacted (see details in SI).' and included a general structure of internal alkenes, noting 'no convn.' in Table 2. This reviewer checked the 'details in SI', and only one internal E-alkene was shown in 'Substrate scope of failed alkenes/hydrazines' on page 6 in SI. It might be too arbitrary to conclude that with the negative results obtained with 1 or several tries. More importantly, probably, this is scientifically and logically incorrect to conclude 'xx do not work' based on failed examples.

Thanks for the reviewer's suggestion. We have adjusted the description and the substrate scope in the manuscript as suggested.

REVIEWERS' COMMENTS

Reviewer #2 (Remarks to the Author):

My comments have been sufficiently addressed. I endorse its acceptance.